# Prospective Evaluation of 3 T MRI Effect on Residual Hearing Function of Cochlea Implantees

**DOI:** 10.3390/brainsci12101406

**Published:** 2022-10-19

**Authors:** Theda Eichler, Ahmed Ibrahim, Conrad Riemann, Lars Uwe Scholtz, Hans Björn Gehl, Peter Goon, Holger Sudhoff, Ingo Todt

**Affiliations:** 1Department of Otolaryngology, Head and Neck Surgery, Medical School OWL, Bielefeld University, Klinikum Bielefeld, 33502 Bielefeld, Germany; 2Department of Radiology, Medical School OWL, Bielefeld University, Klinikum Bielefeld, 33502 Bielefeld, Germany; 3Division of Dermatology, National University Health System, Department of Medicine, Yong Loo Lin School of Medicine, National University of Singapore, Singapore 119228, Singapore

**Keywords:** MRI, residual hearing function, hair cells, cochlea implant

## Abstract

Introduction: The approval process for MRI safety of implants includes physical observations and an experimental evaluation in artificial settings to simulate the in vivo effect. This contains the observation of temperature changes and artificial current generation by the magnetic field. From these findings, the safety of an implant and its effect on the patient can be estimated. MRI safety is based on an in vivo evaluation of adverse events after the approval process, but an actual analysis of the effect on different tissues is not followed. The effect of MRI scanning in cochlea implantees on their residual hearing as the correlate of the hair cell function is so far unknown, therefore the aim of the present study was to observe the effect of 3 T MRI on the residual hearing of cochlea implantees. Material and Methods: In this prospective study, we performed a 3 T MRI T2 2D MS Drive sequence in eight cochlea-implanted ears. Before and after the MRI scan, a bone conduction pure tone audiogram (BC PTA) was performed. All cochlea implantees had a pre-scanning threshold of low frequency residual hearing between 20 dB and 65 dB. Results: Low frequency mean residual hearing was not affected by the 3 T T2 2D MS Drive sequence. We observed a pre-scanning threshold at 250 Hz of 42.9 (SD 3.9) dB and for 500 Hz 57.1 (SD 6.4) dB. Post-scanning BC PTA was for 250 Hz 42.1 (SD 3.9) dB and for 500 Hz 57.1 (SD 5.7) dB. Conclusion: 3 T MRI scanning has no significant functional effect on the hair cells in cochlea implantees in low frequencies with a T2 2D MS Drive sequence.

## 1. Introduction

Cochlea implantation is the treatment of choice for deaf individuals and patients with residual hearing and insufficient speech perception with hearing aids.

The behavioral safety and security of implantable devices in humans, during and after MRI scanning, is of central importance for the usability of such medical devices. MRI is an important clinical tool, and its growing use makes unfavorable MRI behavior a factor which significantly limits the usability of a medical device. MRI behaviour of cochlea implant devices developed from a categorization as “MRI- un-safe” [1] to the recommendation of head wraps [2] with complications like demagnetiza-tion, pain and dislocated magnets [3,4]. A second generation of MRI specifically designed magnets (e.g., diametral rotation bipolar, Medel Synchrony 1, Innsbruck, and Medel) which solved the initial problems [5], was then followed by a third generation of magnets with improved attachment properties due to changes in the magnet metallurgy (Vector Magnet, Medel Synchrony 2, Innsbruck, Austria).

The underlying approval process is complex. The approval for MRI safety of active implantable devices involve numerous tests which evaluate different parameters of the implant (ISO TS 10974:2018 (assessment of the safety of magnetic resonance imaging for patients with an active implantable medical device), but the in vivo effects on the surrounding tissue, and especially sensitive structures like hair cells, are so far unknown.

The approval process includes observations of temperature changes and artificially induced electric currents by the applied magnetic field. These observations before approval are based on a structured experimental setting, evaluating theoretical borders in which an occurrence of the effect of the surrounding tissue function is unlikely. A period of clinical observation is not part of the approval process.

The inserted cochlea implant electrode is very close to the neural structures of the cochlea, and the distance to the sensory hair cells of the cochlea could be less than a millimeter. In comparison to the pacemaker field, which has a comparable approval process, this relationship between the electrode and surrounding tissue is substantially different because the pacemaker field is without sensory hair cells. This constellation in the inner ear is of great interest, because of the evolving indications of implants in patients with residual hearing. Therefore, the effect of MRI on the hair cell function of cochlea implantees need to be considered, since it addresses the potential risk of hearing loss causation by the MRI scan.

The aim of the present study was to observe the effect of 3 T MRI on the postoperative substantial residual hearing of cochlea implantees.

## 2. Material and Methods

In this prospective study, we observed 8 cochlea-implanted ears operated on in 2021/2022 (4 female, 4 male; mean age 60.1 years old). The length of insertion was up to 28 mm (Medel Flex 28, Innsbruck, Austria; SlimJ and HFMS, Advanced Bionics, Stäfa, Swiss; 622, Cochlear, Melbourne, Australia). All patients had a substantial residual hearing at 250 and 500 Hz between 20 dB and 65 dB, postoperatively.

All patients included in the study had a bone conductive pure tone audiogram (BC PTA) before and after the MRI scanning. The period between the MRI scanning and secondary BC PTA was between 5 min and 4 days (mean 1.9 days). During the scan, all patients wore a protective headphone to decrease the risk of a noise-induced hearing loss. A frequency threshold comparison was performed for 250 Hz and 500 Hz since it was present for all patients. Single patients had additional persisting frequency thresholds which were not included for a better comparability.

The MRI is part of our routine postoperative protocol for the evaluation of the intra-cochlea electrode position [6].

MRI: Achieva, Philips Medical System, Best, The Netherland

Sequence: T2 2D Drive MS, voxel size 0.3 × 0.3 × 0.9; FOV 150 × 150, TE 100 ms TR 3000, TSE tact 17, multi shot, flip angle 90°, refocus control 120, metric 512; NSA 5, foldover: AP, SAR: <1.6 W/kg whole body

Estimation of insertional depth in angle was performed by a NEW TOM VGI, Verona, Italy.

Parameter: FOV 15 × 15 cm, 10.48–20.52 mAS, KV 110, 360° followed by a 2D and 3D reconstruction at an external workstation (NNT, main station).

Statistics: we used the Wilcoxon matched pairs signed-rank test (SPSS 24, IBM, Armonk, NY, USA).

Procedures conformed to the World Medical Association’s Declaration of Helsinki and were approved by the University of Münster Faculty of Medicine and Health Sciences Research Ethics Committee (reference: 135-f-S-Amendment). All participants gave their written informed consent.

## 3. Results

We observed no significant difference between the pre- and post-scanning BC PTA at 250 and 500 Hz. (Table 1) (2 tailed *p* > 0.9999 for both). The mean pre-scanning BC PTA was for 250 Hz 42.9 (SD 3.9) dB and for 500 Hz 57.1 (SD 6.4) dB. The post-scanning BC PTA was for 250 Hz 42.1 (SD 3.9) dB and for 500 Hz 57.1 (SD 5.7) dB. A loss of a frequency response was not observed. The mean insertional angle was 442.5°, ranging from 370° to 560°.

## 4. Discussion

Cochlea implantation is the treatment of choice for deaf individuals and patients with residual hearing and insufficient speech perception with hearing aids.

The MRI behavior of medical implants is of a high importance for their clinical usability. An MRI-unsafe device has only a limited clinical usability since there is the widespread use of MRI scanners in routine practice [7], and they are crucial for diagnostic pathways in the clinics.

Cochlea implant device improvements have addressed the problematic MRI behavior (after a period of developmental steps) from MRI-unsafe [1] over-recommended specific procedures (head wrapping) [2] to technical solutions, based on the magnetic design or screwing of the implant [8,9,10].

The approval process of the devices is complex and contains numerous tests based on a standardized protocol.

The ISO TS 10974:2018 (assessment of the safety of magnetic resonance imaging for patients with an active implantable medical device) protocol evaluates multiple factors (Table 2).

One limitation of the approval process for cochlea implants is a lack of an in vivo evaluation of the surrounding tissue, especially the labyrinthine hair cell function. These cells (even without a cochlea implant) are known to be affected temporarily by MRI scans and can be stimulated by the magnetic field [11,12].

While MRI scans without a cochlea implant have been shown to affect the hearing function, relating to the scanner-generated noise exposure at 1.5 T [13], the additional use of fitted protective headphones limits or excludes the effect on the hearing function [12].

We observed in our study no statistically significant BC threshold difference between the mean pre- and post-scanning observation, focusing on a short time period of up to four days (Table 1). By this focus on a short time period, other factors causing a loss of residual hearing can be minimized.

Hair cells are known to be temperature sensitive, and their function is temporary affected by a temperature increase of more than 1.4 °C [14] in humans, and they become irreversibly changed at the temperature of 41–42 °C [15] in rats. In our in vivo study, such temperature changes do not seem to be generated close to the hair cells. MRI scans also do not seem to cause harmful current inductions or a detrimental change in inner ear homoeostasis.

This finding is of great importance since not only are the number of MRI scans is increasing [7], but the number of cochlea implantees with residual hearing is also rising [16]. These increases are related to a shift in the indications for cochlea implantation into the group of patients with residual hearing.

A physical overlap of the residual hearing region in our group (to 500 Hz) and electrode insertion depth (mean 442.5°, max. 560°) can be assumed to be present based on the observations of Stakhovskaya [17] for most of our patients (Table 1). Stakhovskaya and colleagues observed the frequency of a 500 Hz location of the organ of Corti to be under 450°. The mean frequency at 450° is 601 Hz, with a size-dependent frequency range between 531 Hz and 699 Hz. This may cause different temperature changes around the hair cells related to the cochlea size [18] and electrode length, depending on the proximity, than in our study. Figure 1a,b (Pat. 6) shows an unchanged BC threshold of up to 1 kHz. Pat. 7 has a Flex 28 electrode with an insertional depth of 560°. Therefore, for these patients, a negative effect on their hearing function based on a physical overlapping can be excluded. Since the intra-cochlea current is known to spread widely during a stimulation, distance can be excluded as an explanation for the lack of effect on the hair cell function.

A further limitation of our study is the use of only a single MRI sequence (T2 2D MS Drive). This sequence is commonly used in our department for a postoperative evaluation [6] and is therefore the first step into this field. Different MRI-specific absorption rate (SAR) values of different sequences should be investigated in future studies. The SAR value for our used sequence is max 1.6 W/kg and therefore an orientation for the use of other sequences.

## 5. Conclusions

Three T MRI scanning has no significant negative effect on the hair cell function in cochlea implantees in low frequencies with a T2 2D MS Drive sequence.

## Figures and Tables

**Figure 1 brainsci-12-01406-f001:**
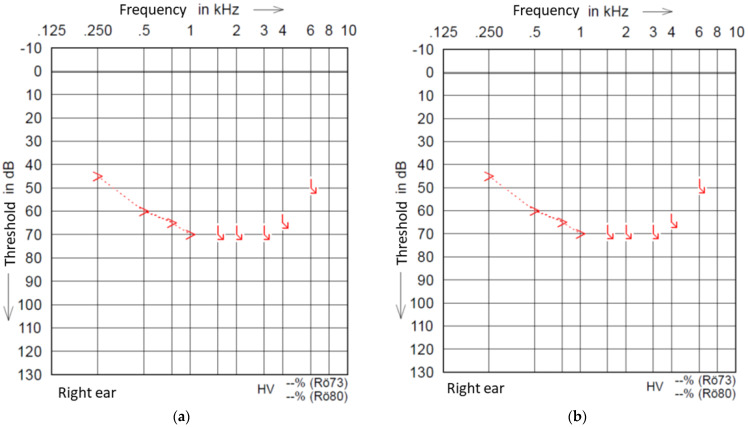
(**a**) Example pre-scanning PTA of Pat. 6. (**b**) Example post-scanning PTA of Pat. 6.

**Table 1 brainsci-12-01406-t001:** Legend Table 1: individual BC PTA data pre and post MRI scanning, insertional angle in ° and the timespan between MRI scan and post-scanning BC PTA in days.

Pat.	Insertional °	250 Hz Pre	500 Hz Pre	250 Hz Post	500 Hz Post	Timespan/Days
1	430	35	55	45	60	4
2	450	45	45	40	45	4
3	450	45	60	45	60	1
4	430	45	60	35	60	1
5	420	40	55	40	55	1
6	430	45	60	45	60	0
7	560	45	65	45	60	1
8	370	5	15	10	15	3

**Table 2 brainsci-12-01406-t002:** Legend Table 2: ISO 10974 testing for assessment of MRI safety of active implantable medical devices. B0—static effect; RF—radio frequency-induced effects; and gradient-induced effects generated by the change in magnitude and direction of the magnetic field.

B0-Induced Force	Gradient-Induced Vibration, Device Malfunction
B0-induced torque	Gradient-induced vibration, tissue damage
B0-induced demagnetization	Gradient-induced unintended stimulation
B0-induced magnet dislocation	Gradient-induced case heating
B0-induced malfunction	Gradient-induced malfunction, radiated
Gradient-induced malfunction, injected	RF-induced electrode heating
RF-induced unintended stimulations	RF-induced malfunction
RF-induced case heating, radiated	Combined field test
RF-induced case heating injected	Image artefact test

## Data Availability

The data are available from the corresponding author upon request.

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
