# Peer review of "Prospective Evaluation of 3 T MRI Effect on Residual Hearing Function of Cochlea Implantees"

_brainsci, 2022, doi:10.3390/brainsci12101406_

Round 1

Reviewer 1 Report

Thank you for your great work and well written manuscript.

1- However, I have a question: in the materials and methods section you mentioned "The period between MRI scanning and secondary BC PTA was between 5 min and 4 days year (mean 1.9 days)" Is this time ( minimum 5 min, maximum 4 days) sufficient to evaluate hair cell function changes? In other words, is it possible the effect of temperature changes during MRI scanning occurs more lately?

2- What do you mean about the word year in this sentence" The period between MRI scanning and secondary BC PTA was between 5 min and 4 days year (mean 1.9 days) , line 72? 

Author Response

Dear reviewer,

thank you for the raised questions.

1)This is a very good question. We discussed about that too and found that only an early observation can show an purely MRI related effect. As longer after the insertion you wait as higher is the probability that other factors contribute or mix to the occurence of hearing loss.

2) This is a typo which is changed.

Thank you for your helpful comments !

Reviewer 2 Report

The authors measured bone conduction hearing at 250 and 500 hz before and after a 3T specific sequence MR session in 8 CI pts operated with EAS technique. Hearing thresholds were preserved after the MR session indicating that this MR sequence is harmless to the preoperatively functioning cochlear hair cells.

The study is useful and methods are sound, conclusions should be valid. Limitations well discussed. 

A typo error line 72 - year must be an error.

the beginning of the discussion has a lot of repetition - ok with me.

Author Response

Dear reviewer,

thank you for the positive comments. The typo is changed.